# A Study on the Change in Properties by Using an Additive with Water-Soluble Rubber-Asphalt-Based Waterproof Coating Materials

**Sung-hwan Yoon [1], Neung-hoe Heo [1], Ki-won An [2] and Sang-keun Oh [3,*]**

[1] SEWON Waterproof Co., Ltd., 301-7, Pokpo-ro, Hwado-eup, Namyangju-si, Gyeonggi-do 02196, Korea; ysh9627@gmail.com (S.-h.Y.); nhwing@nate.com (N.-h.H.)

[2] Institute of Construction Technology, Seoul National University of Science & Technology, 232 Gongneung-ro, Nowon-gu, Seoul 01811, Korea; ankiwon@seoultech.ac.kr

[3] School of Architecture, Seoul National University of Science & Technology, 232 Gongneung-ro, Nowon-gu, Seoul 01811, Korea

* Correspondence: ohsang@seoultech.ac.kr; Tel.: +82-2-970-6559

**Abstract:** Water-soluble rubber-asphalt-based waterproof coating materials are mainly used for the purpose of preventing leakage from the inside of a structure to the outside or to the lower floor. In this study, water-soluble rubber-asphalt-based waterproof coating materials were prepared based on different mixture ratios to develop a new method to improve the physical properties of the materials. In order to improve the overall physical properties, additives were added to the coating material and analyzed for the change in performance, and the analyzed additives were used in combination to improve the physical properties of waterproofing materials. In particular, in the quality standards, 'Attachment performance and fatigue resistance performance are the performance required for the outside of the structure', but in the rubber asphalt category, the value of the adhesion performance quality standard was excluded, so it was limited to external use as a waterproofing material. The degree of improvement was compared and analyzed according to the quality standards of the Korean Industrial Standard KS F 3211-'15' Paint Waterproofing Materials for Construction. Therefore, in this study, a total of 27 complex additive tests were conducted, and among them, basic property evaluation was performed on 14 items, excluding 13 items that had defects in the drying process. As a result, in the test conducted with complex additives, about 98% to 104% of the quality standard was obtained, and it was judged that stable waterproofing performance would appear if additional quality improvements were made based on this combination in the future.

**Keywords:** soluble rubber asphalt; waterproof coating; additive; property enhancement

## 1. Introduction

Water-soluble rubber asphalt coating waterproofing materials are mainly used in domestic building structures and are used for the purpose of preventing water leakage in areas where water is used indoors, such as shower rooms, toilets, verandas, and kitchens. Looking at the construction process, a water-soluble rubber-asphalt-based coating waterproofing material is installed on the surface to which the waterproofing material is applied, and a protective mortar with a gradient that allows water to flow is installed on the top [1–4]. Since these water-soluble rubber-asphalt-based paint film waterproofing materials are mainly applied to non-structural parts, the minimum tensile strength is set as the quality standard even within the Korean Industrial Standards. Accordingly, water-soluble rubber-asphalt-based coating film waterproofing materials produced in Korea have the characteristics of low tensile strength and high elongation [5,6], as shown in Figure 1. Below, it was confirmed that the quality standards for water-soluble rubber-asphalt-based waterproofing materials were set because the construction site of the waterproofing layer

was less damaged by external forces, but since the bathroom balcony is a non-structural part, it requires a minimum waterproofing performance [7–11].

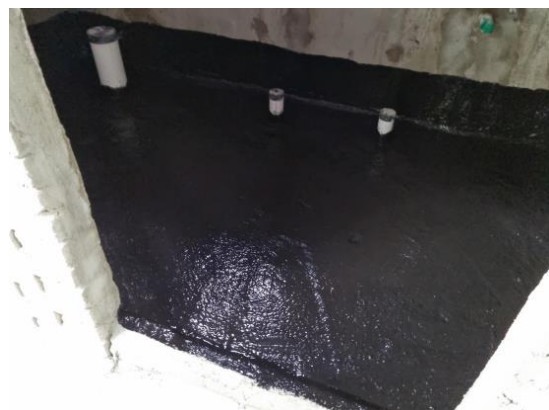 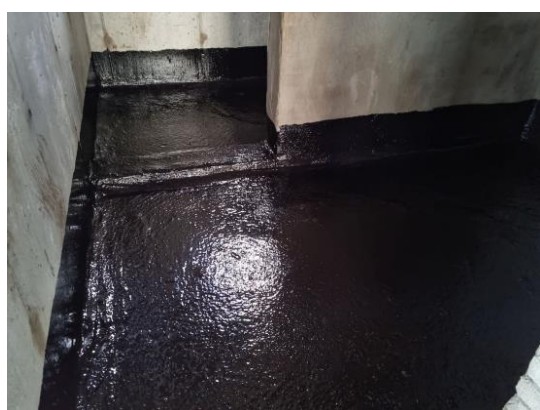

**Figure 1.** Water-soluble rubber-asphalt-based waterproof coating installation on floor.

In particular, unlike the past, which consisted of single or low floors, modern buildings are taller and larger, so the structure's vibrations and behaviors are more significant [12,13]. The low physical properties of the waterproofing layer cause simultaneous rupture of the waterproofing layer along with cracks in the slab, which causes inter-layer leakage and causes disputes between users of the structure [14–17]. Therefore, the reality is that it is necessary to develop a waterproofing material with properties superior to that of the current rubber-asphalt-based waterproofing material. Therefore, in this study, a method for improving the physical properties of water-soluble rubber-asphalt-based waterproofing materials is studied, and in order to improve the overall physical properties, a single additive is added to collect and analyze the change in performance, and the analyzed additives are used in combination to improve the physical properties of waterproofing materials [18–21]. The degree of improvement is compared and analyzed according to the quality standards of the Korean Industrial Standard KS F 3211-'15' Paint Waterproofing Materials for Construction.

On the topic of economic applicability, the purpose of this study is to compare the degree of improvement in physical properties according to additives. Additives were selected and research was conducted within the range that did not exceed the production cost of other waterproofing materials (about 80–90%).

## 2. Theoretical Considerations

*2.1. Confirmation of the Basic Mixing Ratio and Physical Properties of Materials Used*

2.1.1. Basic Mixing Ratio

Table 1 below shows the basic mixing ratio of water-soluble rubber-asphalt-based waterproofing materials used in this study.

**Table 1.** Mixing ratio of water-soluble rubber-asphalt-based waterproof coating.

| Material Type | Ratio (%) |
| --- | --- |
| Emulsifier | 0.1–5.0 |
| Latex | 20–50 |
| Asphalt | 30–70 |
| Water | 1–10 |
| Others | 1.0–5.0 |

2.1.2. Confirmation of the Basic Properties of Materials Used

The water-soluble rubber-asphalt-based coating waterproofing material used in this study was used in accordance with the rubber-asphalt-based quality standards of the

Korean Industrial Standard KS F 3211-'15' 'Construction Coating Waterproofing Materials'. Table 2 below shows the performance and quality standards for tensile strength, elongation, tear performance, solid content, and viscosity (hereinafter, basic physical properties) and the degree of improvement compared to the quality standards.

**Table 2.** Water-soluble rubber-asphalt-based waterproof coating performance and quality standard.

| Quality Standard Criteria | | Performance Result | Quality Standard | Performance Enhancement Compared to Quality Standard |
|---|---|---|---|---|
| Tensile Performance | Tensile Strength (N/mm$^2$) | 0.33 | More than 0.3 | 110% |
| | Elongation (%) | 1352 | More than 600 | 225% |
| Shear Strength (N/mm) | | 3.21 | More than 2.9 | 111% |
| Solidity Ratio (%) | | 73.33 (Standard value = 75) | ±3% compared to standard | 1.7% |
| Viscosity (mPa/s) (LV 3 Spindle 20 rpm, 23 °C) | | 522 | N/A | N/A |

As a result of checking the performance according to the material used, it was confirmed that the tensile strength was improved by about 110% compared to the quality standard, and the elongation was improved by about 225% compared to the quality standard. In addition, the tear strength was confirmed to be improved by about 111% compared to the quality standard, and the solid content was confirmed to be 1.7% decreased compared to the indicated value. Lastly, in the case of viscosity, there is no quality standard, but since it is possible to secure fine grinding, pinholes, and flowability when placing the test piece by maintaining the appropriate viscosity, in this study, the viscosity test value was measured to confirm the workability.

## 2.2. Additive Selection

For the selection of additives, additives that can improve the tensile performance of the material, maintain an appropriate elongation, and improve the tear strength within the range of maintaining an appropriate viscosity for workability were selected. The selected additives were as follows.

### 2.2.1. Silane Coupling Agent

The silane coupling agent used in this study is mainly used for modifying materials and resins or for bonding and has the characteristic of compounding organic and inorganic materials. The structure of the silane coupling agent is characterized by having at least two types of functional groups with different reactivity in one molecule and is divided into a reactive group that chemically bonds with an organic material and a reactive group that chemically bonds with an inorganic material, as shown in Figure 2.

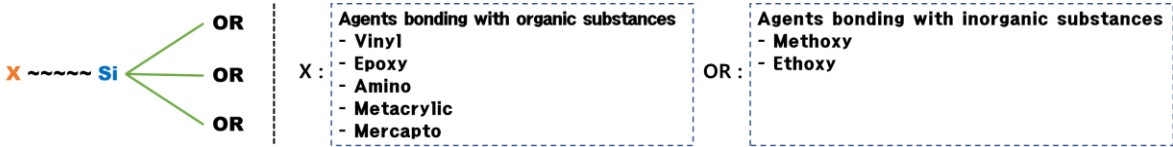

**Figure 2.** Sealant coupling agent structure.

The silane coupling agent has both organic and inorganic functional groups, so it acts as a crosslinking agent capable of chemical bonding of both organic and inorganic materials; this reaction improves the tensile performance of rubber, synthetic resin, paint, and the adhesive. In particular, it has the property of increasing the bonding strength of

the materials by integrating different materials, such as organic materials and inorganic materials. In addition, the alkoxysilyl group is hydrolyzed by moisture or moisture in the material to be synthesized, and it is called a silanol group (Si-OH). These silanol groups cause a condensation reaction with the inorganic surface and induce Si-O-M bonding by the condensation reaction, as shown in Figure 3.

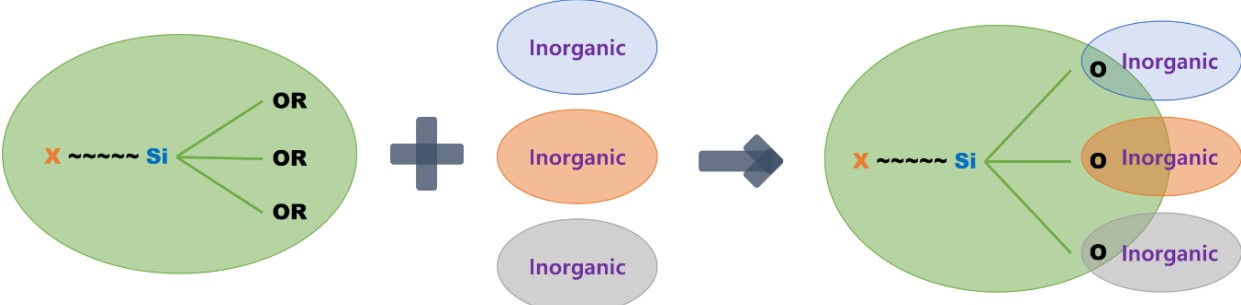

**Figure 3.** Silane coupling agent inorganic coupling reaction.

In addition, in the case of another reactive group, X, it is possible to chemically combine inorganic substances and organic substances by combining with organic substances, as shown in Figure 4.

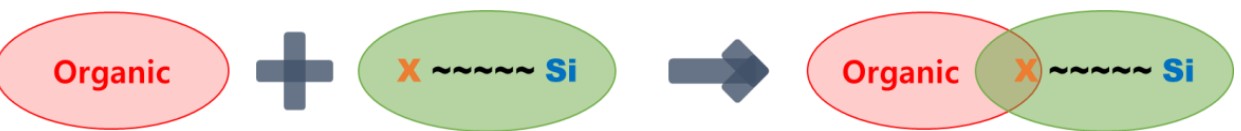

**Figure 4.** Silane coupling agent organic coupling reaction.

The overall coupling reaction of the silane coupling agent is shown in Figure 5.

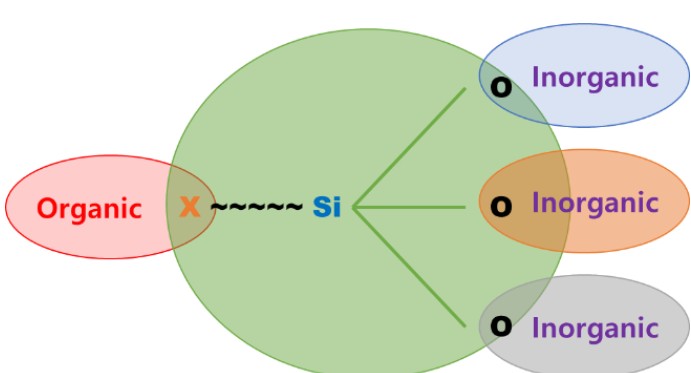

**Figure 5.** Total coupling reaction of silane coupling agent.

Table 3 below shows the types of silane coupling agents used in Korea.

Therefore, in this study, '3-Glycidyloxy Propyl Trimethoxy Silane' was used as an additive for water-soluble rubber asphalt coating film.

In this study, the mixing method of the silane coupling agent was tested by simultaneously adding and mixing the silane coupling agent when mixing the organic resin and the inorganic material (filler), which was mixed through the integral blend additive method. In the method, the silanol group (Si-OH) generated during hydrolysis of the silane coupling agent is combined with an inorganic additive, and the reactor (X) is combined with an organic raw material to improve physical performance.

**Table 3.** Types of silane coupling agents.

| Chemical Name | Minimum Area (m²/g) | CAS No. |
|---|---|---|
| Vinyl Trichloro Silane | 480 | 75-94-5 |
| Vinyl Trimethoxy Silane | 515 | 2768-02-7 |
| Vinyl Ris(2-Methoxy Ethoxy) Silane | 410 | 1067-53-4 |
| 2-(3,4 Epoxycyclohexyl) Ethyltrimethoxy Silane | 278 | 3388-04-3 |
| 3-Glycidyloxy Propyl Trimethoxy Silane | 317 | 2530-83-8 |
| 3-Glycidyloxy Propyl Methyl Diethoxy Silane | 330 | 2897-60-1 |
| 3-Glycidyloxy Propyl Triethoxy Silane | 356 | 2602-34-8 |
| 3-Methacryloxy Propyl Methyl Dimethoxy Silane | 280 | 14513-34-9 |
| 3-Methacryloxy Propyl Trimethoxy Silane | 314 | 2530-85-0 |
| 3-Methacryloxy Propyl Nethyl Diethoxy Silane | 300 | 65100-04-1 |
| N-2(Amino Ethyl)2-Amino Propyl methyl dimethoxy Silane | 380 | 3069-29-2 |
| N-2(Amino Ethyl)2-Amino Propyl Trimethoxy Silane | 351 | 1760-24-3 |
| N-2(Amino Ethyl)2-Amino Propyl Triethoxy Silane | 295 | 5089-72-5 |
| 3-Amino Propyl Trimethoxy Silane | 436 | 13822-56-5 |
| 3-Amino Propyl Triethoxy Silane | 353 | 919-30-2 |
| N-Phenyl-3-Amino Propyl Trinethoxy Silane | 307 | 3068-76-6 |
| 3-Chloro Propyl Trimethoxy Silane | 393 | 2530-87-2 |
| 3-Mercapto Propyl Trimethoxy Silane | 398 | 4420-74-0 |

### 2.2.2. Filler

Fillers are generally used to improve the strength or properties of materials and are defined as inactive substances added to plastics in the production process. Due to the recent rapid growth of the material field, the uses of fillers are being diversified, and as shown in Figure 6, depending on the use of the material, the shape of the material (plate-like, powder-type, or fiber-type) and its role are also expanding.

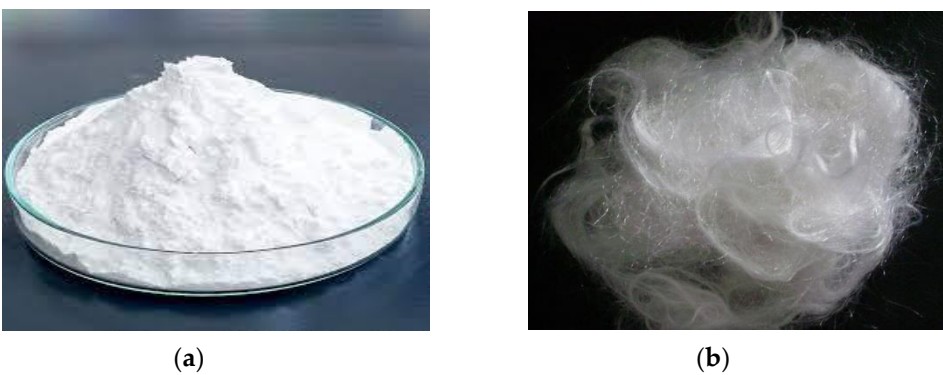

| (**a**) | (**b**) |
|---|---|

**Figure 6.** Types of fillers. (**a**) Fine powdered filler; (**b**) fiber-type filler.

Inorganic fillers are used for improving the performance of plastics and the like to improve the properties of raw materials. Therefore, they are not simply for the purpose of increasing the volume but are used as a substitute for expensive plastics, such as engineering plastics and ABS plastics such as polypropylene (PP) filled with talc, thereby satisfying economic requirements. In addition, they play an important role in improving the mechanical and thermal properties of rubber and plastics to improve the performance of materials.

### 2.2.3. EVA (Ethylene Vinyl Acetate)

It is a copolymer resin of ethylene and vinyl acetate. Unlike general PVC, it is light, soft, easy to process, and easily deformed when heated, making it one of the most used materials in various fields. In particular, the density, flexibility, etc., increase according to the content of vinyl acetate, and the elasticity, heat bonding temperature, durability,

and penetrating power are different. In addition, it has greater transparency and adhesion compared to LDPE, excellent elasticity, low-temperature heat-sealing properties, and is economically superior to other polar polymers, such as polyurethane.

The molecular structure of EVA is shown in Figure 7.

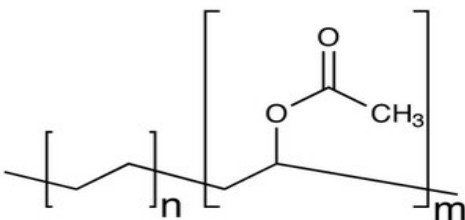

**Figure 7.** EVA's molecular structure.

EVA forms a polymer main chain by randomly mixing LDPE (low-density polyethylene) with vinyl acetate. When the content of vinyl acetate is increased, the strength, crystallinity, melting point, and chemical durability decrease, and impact strength, transparency, and friction coefficient increase. In the synthesis process for EVA, ethylene and vinyl acetate are produced in a continuous process through free-radical addition polymerization in a high-pressure reactor. This is a chain-growth polymerization method in which the initiator is broken to form free radicals and continuously reacts with excessively added double-bond monomers (vinyl monomers). Therefore, in this study, it was used as an additive to improve the elongation of the waterproofing material in the final formulation of the material.

### 2.3. Research Flow

According to the purpose of this study, the mixing ratio and basic physical properties of the water-soluble rubber-asphalt-based coating waterproofing material confirmed in this theoretical review were checked, and the silane coupling agent and filler (nylon fiber) and properties for EVA were confirmed. Accordingly, in the following Figure 8, the flow of the study was established.

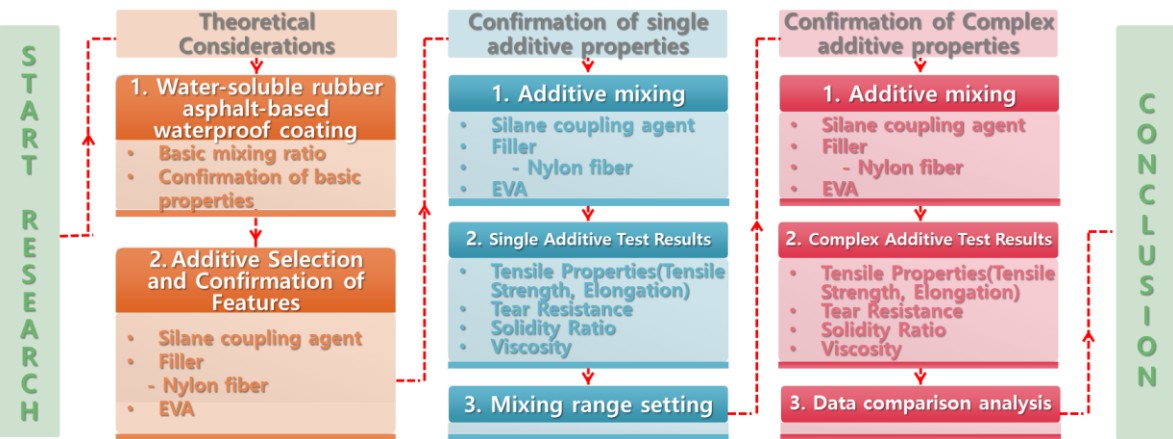

**Figure 8.** Research flow chart.

## 3. Confirmation of Single Additive Properties

### 3.1. Test Plan and Method

In this study, research was conducted to improve the tensile performance (tensile strength, elongation), tearing performance, viscosity, and solid content of water-soluble rubber asphalt coating waterproofing materials according to the use of additives—we

planned to evaluate the degree of improvement in quality compared with the performance items of six types of waterproofing materials for exposure (Table 4).

**Table 4.** KS F 3211—quality standards for basic physical properties [13].

| Items | | Types | | | | | |
|---|---|---|---|---|---|---|---|
| | | Urethane Rubber Type 1 | Acrylic Rubber | Chloroprene Rubber | Silicone Rubber | Urethane Rubber Type 2 | Rubber Asphalt |
| Tensile Performance | Tensile Strength (N/mm$^2$) | More than 2.5 | More than 1.5 | More than 1.5 | More than 0.5 | More than 2.0 | More than 0.3 |
| | Elongation (%) | More than 450 | More than 300 | More than 450 | More than 600 | More than 550 | More than 600 |
| Tear-Resistance Performance | Tear-Resistance Strength (N/mm) | More than 14.7 | More than 6.9 | More than 14.7 | More than 2.9 | More than 12.8 | More than 2.9 |
| Solidity Ratio (%) | | Increase from standard value ± 3 | | | | | |

### 3.1.1. Tensile Performance (Tensile Strength, Elongation) and Tearing Performance

The tensile performance, tear performance, and solid content of the water-soluble rubber asphalt waterproofing material were prepared according to the Korean Industrial Standard KS F 3211-'15' method for producing rubber-asphalt-based test pieces in the construction coating film waterproofing material, and the test was carried out. In addition, the degree of physical property improvement for the tensile performance (tensile strength, elongation) and tearing performance of the materials used in this study was analyzed by comparing the physical properties of urethane rubber type 1, acrylic rubber type, chloroprene rubber type, and silicone rubber type.

### 3.1.2. Solidity

The solid content of the water-soluble rubber asphalt waterproofing material was tested by applying the Korean Industrial Standard KS F 3211-'15' test method for rubber asphalt in the coating waterproofing material for mutatis mutandis construction. In this study, the measurement of solid content was meaningful to predict the thickness reduction in the coating film waterproofing material that occurs during the curing process of the material as it exists as the indicated value of the material even within the quality standards.

### 3.1.3. Viscosity

The viscosity of the water-soluble rubber asphalt coating film waterproofing material was tested by applying the Korean Industrial Standard KS M ISO 2555-'17' plastics—liquid, suspended, or dispersed resin—the test method for apparent viscosity by the mutatis mutandis Brookfield method. Since there is no separate quality standard for the measurement of viscosity in this study, it was used for measurement to confirm the workability according to the spread or agglomeration of the material during the construction process.

### 3.2. Setting the Single Additive's Mixing Ratio

In this study, the addition range for each additive was set to confirm the changes in physical properties when the water-soluble rubber-asphalt-based waterproofing material and the selected additive were used alone. The amount of each additive added is shown in Table 5 below.

**Table 5.** Single additive mixture ratio.

| Item | Mixture Ratio | Increment | Input |
|---|---|---|---|
| Silane coupling agent | Min 0.0%–Max 1.5% | 0.3% unit increase | 0.3%, 0.6%, 0.9%, 1.2%, 1.5% |
| Nylon fibre | Min 0.0%–Max 2.5% | 0.5% unit increase | 0.5%, 1.0%, 1.5%, 2.0%, 2.5% |
| EVA | Min 0.0%–Max 2.5% | 0.5% unit increase | 0.5%, 1.0%, 1.5%, 2.0%, 2.5% |

As for the setting method, a water-soluble rubber-asphalt-based coating waterproofing material, whose properties were confirmed in theoretical consideration, was selected as the basic material in Table 6, and a silane coupling agent (0.3, 0.6, 0.9, 1.2, 1.5%), nylon fiber (0.5, 1.0, 1.5, 2.0, 2.5%), or EVA (0.5, 1.0, 1.5, 2.0, 2.5%) were, respectively, blended to confirm the physical properties for a total of 15 test groups.

**Table 6.** The physical properties of water-soluble rubber-asphalt-based waterproofing materials according to the type and content of additives.

| Item (Code) | Mixture Ratio | | | Test Results | | | | |
| | Soluble Rubber Asphalt Waterproof Coating | | | Tensile Strength (N/mm$^2$) | Elongation (%) | Tear Resistance (N/mm$^2$) | Solidity Ratio (%) | Viscosity (mPa·s) |
| | Silane Coupling Agent (SCA) | Nylon Fibre (NF) | EVA | | | | | |
|---|---|---|---|---|---|---|---|---|
| STD | - | - | - | 0.33 | 1352 | 3.21 | 73.33 | 522 |
| SCA 0.3 | 0.3% | - | - | 0.57 | 1044 | 5.31 | 79.33 | 726 |
| SCA 0.6 | 0.6% | - | - | 0.63 | 997 | 6.34 | 80.67 | 1297 |
| SCA 0.9 | 0.9% | - | - | 0.70 | 906 | 7.02 | 82.67 | 1840 |
| SCA 1.2 | 1.2% | - | - | 0.78 | 832 | 7.67 | 84.00 | 2715 |
| SCA 1.5 | 1.5% | - | - | 0.81 | 711 | 8.24 | 85.33 | 3313 |
| NF 0.5 | - | 0.5% | - | 0.68 | 704 | 6.25 | 85.33 | 1327 |
| NF 1.0 | - | 1.0% | - | 1.12 | 412 | 8.56 | 87.33 | 2034 |
| NF 1.5 | - | 1.5% | - | 1.54 | 318 | 11.23 | 88.67 | 2877 |
| NF 2.0 | - | 2.0% | - | 1.97 | 181 | 14.54 | 90.67 | 3561 |
| NF 2.5 | - | 2.5% | - | 2.43 | 83 | 17.78 | 91.33 | 4322 |
| EVA 0.5 | - | - | 0.5% | 0.30 | 1418 | 2.51 | 77.33 | 690 |
| EVA 1.0 | - | - | 1.0% | 0.28 | 1733 | 2.18 | 80.00 | 1221 |
| EVA 1.5 | - | - | 1.5% | 0.25 | 2154 | 1.77 | 82.67 | 1765 |
| EVA 2.0 | - | - | 2.0% | 0.24 | 2214 | 1.72 | 84.67 | 2113 |
| EVA 2.5 | - | - | 2.5% | 0.24 | 2291 | 1.73 | 86.00 | 2468 |

### 3.3. Single Additive Test Results

Changes in physical properties according to the additives analyzed based on the above results are as follows.

### 3.3.1. Silane Coupling Agent

As shown in Table 7 below, for water-soluble rubber-asphalt-based waterproofing materials whose properties were confirmed by adding only a silane coupling agent, the tensile strength, tear strength, solid content, and viscosity increase as the amount added increases due to the bonding cohesive force of the silane coupling agent. On the contrary, the elongation rate decreased. In the case of solid content, it was confirmed that the increase was about 12%, but in the case of viscosity, a large rate of increase was confirmed, about 635%. It is thought that the increase in physical properties except for elongation occurred as the silane coupling agent improved the bonding force of the material. Therefore, in this study, we judged that the use of a silane coupling agent can potentially improve the strength of the material.

### 3.3.2. Nylon Fiber

As shown in Table 8 below, the water-soluble rubber-asphalt-based paint film waterproofing material, whose properties were checked by adding only nylon fibers, showed that the tensile strength, tear strength, solid content, and viscosity increased as the amount of nylon fiber added increased, especially in the case of viscosity. Compared to the STD, it showed a significant increase of about 828%. Conversely, the elongation rate was decreased, and the elongation rate was significantly reduced to about 6% compared to the STD, which indicates that the nylon fibers dispersed in the water-soluble rubber-asphalt-based waterproofing material are continuously combined with the asphalt to increase the fiber strength. It is estimated that the elongation rate is reflected, and it is considered that the high tensile strength, tear strength, and low elongation rate were caused by this. Therefore, in this study, it is judged that nylon fiber can be used to improve the strength of the material.

**Table 7.** Physical property results according to the amount of silane coupling agent added and increase/decrease rate compared to STD.

| Item (Code) | Tensile Strength | | Elongation | | Tear Resistance | | Solidity Ratio | | Viscosity | |
|---|---|---|---|---|---|---|---|---|---|---|
| | Result (N/mm) | Change from STD (%) | Result (%) | Change from STD (%) | Result (N/mm) | Change from STD (%) | Result (%) | Change from STD (%) | Result (mPa·s) | Change from STD (%) |
| STD | 0.33 | - | 1352 | - | 3.21 | - | 73.33 | - | 522 | - |
| SCA 0.3 | 0.57 | 173 | 1044 | 77 | 5.31 | 165 | 79.33 | 108 | 726 | 139 |
| SCA 0.6 | 0.63 | 191 | 997 | 74 | 6.34 | 198 | 80.67 | 110 | 1297 | 248 |
| SCA 0.9 | 0.7 | 212 | 906 | 67 | 7.02 | 219 | 82.67 | 113 | 1840 | 352 |
| SCA 1.2 | 0.78 | 236 | 832 | 61 | 7.67 | 239 | 84 | 115 | 2715 | 520 |
| SCA 1.5 | 0.81 | 245 | 711 | 53 | 8.24 | 257 | 85.33 | 116 | 3313 | 635 |

**Table 8.** Physical property results according to the amount of nylon fiber input and increase/decrease rate compared to STD.

| Item (Code) | Tensile Strength | | Elongation | | Tear Resistance | | Solidity Ratio | | Viscosity | |
|---|---|---|---|---|---|---|---|---|---|---|
| | Result (N/mm) | Change from STD (%) | Result (%) | Change from STD (%) | Result (N/mm) | Change from STD (%) | Result (%) | Change from STD (%) | Result (mPa·s) | Change from STD (%) |
| STD | 0.33 | - | 1352 | - | 3.21 | - | 73.33 | - | 522 | - |
| NF 0.5 | 0.68 | 206 | 704 | 52 | 6.25 | 195 | 85.33 | 116 | 1327 | 254 |
| NF 1.0 | 1.12 | 339 | 412 | 30 | 8.56 | 267 | 87.33 | 119 | 2034 | 390 |
| NF 1.5 | 1.54 | 467 | 318 | 24 | 11.23 | 350 | 88.67 | 121 | 2877 | 551 |
| NF 2.0 | 1.97 | 597 | 181 | 13 | 14.54 | 453 | 90.67 | 124 | 3561 | 682 |
| NF 2.5 | 2.43 | 736 | 83 | 6 | 17.78 | 554 | 91.33 | 125 | 4322 | 828 |

### 3.3.3. EVA

As shown in Table 9 below, as the amount of EVA added increased, the water-soluble rubber-asphalt-based waterproofing material whose properties were checked by adding only EVA showed that the tensile strength, tear strength, solid content, and viscosity decreased. This is considered to be due to the low crystallinity of EVA (the ratio of the weight of the crystalline part in the total center of the polymer material) and the improvement of the elongation due to elasticity. However, at the addition ratios of 2.0% and 2.5%, as the rates of increase of tensile strength, elongation, and tear strength were similar, only the solid content and viscosity increased. Therefore, in the case of EVA, it is considered that the use of 2.0% or less is required in the composite additive.

**Table 9.** EVA physical property results according to input amount and increase/decrease rate compared to STD.

| Item (Code) | Tensile Strength | | Elongation | | Tear Resistance | | Solidity Ratio | | Viscosity | |
|---|---|---|---|---|---|---|---|---|---|---|
| | Result (N/mm) | Change from STD (%) | Result (%) | Change from STD (%) | Result (N/mm) | Change from STD (%) | Result (%) | Change from STD (%) | Result (mPa·s) | Change from STD (%) |
| STD | 0.33 | - | 1352 | - | 3.21 | - | 73.33 | - | 522 | - |
| EVA 0.5 | 0.30 | 91 | 1418 | 105 | 2.51 | 78 | 64.00 | 105 | 488 | 132 |
| EVA 1.0 | 0.28 | 85 | 1733 | 128 | 2.18 | 68 | 66.00 | 109 | 671 | 234 |
| EVA 1.5 | 0.25 | 76 | 2154 | 159 | 1.77 | 55 | 67.33 | 113 | 851 | 338 |
| EVA 2.0 | 0.24 | 73 | 2214 | 164 | 1.72 | 54 | 71.33 | 115 | 913 | 405 |
| EVA 2.5 | 0.24 | 73 | 2291 | 169 | 1.73 | 54 | 74.67 | 117 | 1094 | 473 |

### 3.4. Analysis of the Single Additive's Test Results

3.4.1. Analysis of Tensile Strength, Elongation Rate, and Tear Strength Results of the Single Additive

Changes in tensile strength, elongation, and tear strength according to the amount of each additive are shown in the following Figures 9–11.

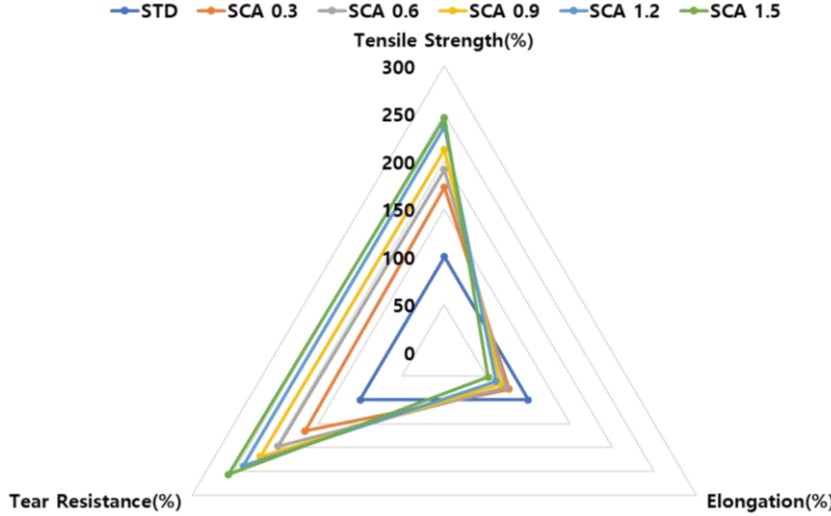

**Figure 9.** Strength increase/decrease rate according to the amount of SCA input.

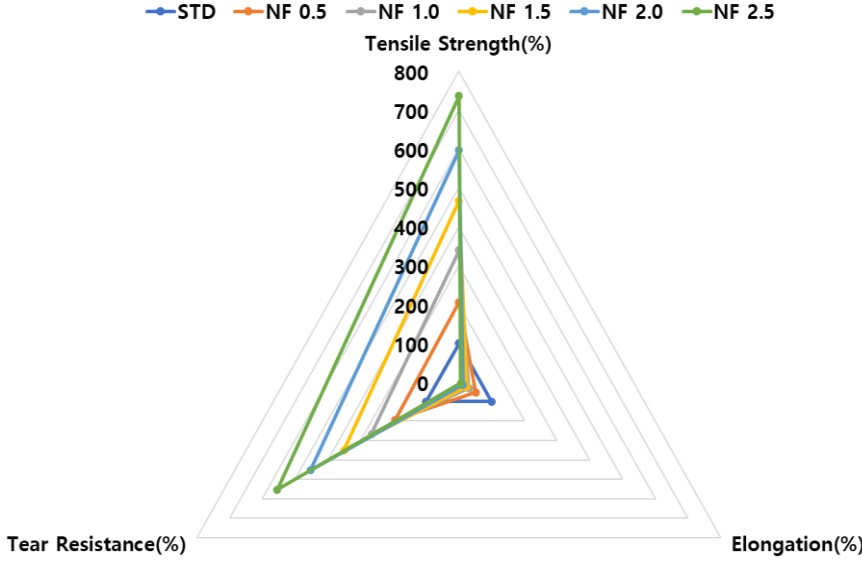

**Figure 10.** Strength increase/decrease rate according to NF input.

As a result of the single additive test, it was confirmed that the tensile strength and tear strength increased by up to about 736% in tensile strength and about 554% in tear strength depending on the amount of silane coupling agent and nylon fiber added. It was confirmed that strength and tear strength were reduced by about 27% and 46%, respectively. In addition, it was confirmed that the elongation was increased to about 169% with the addition of EVA and decreased to about 94% with the addition of the silane coupling agent and nylon fiber.

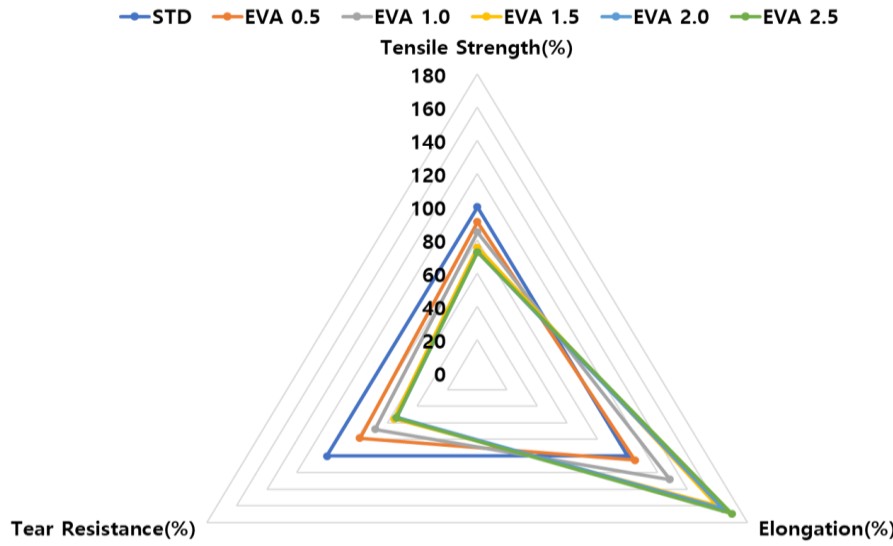

**Figure 11.** Strength increase/decrease rate according to EVA input amount.

3.4.2. Analysis of Solids and Viscosity Results Using a Single Additive

The changes in solid content and viscosity according to the input amount for each additive are shown in the following Figures 12 and 13.

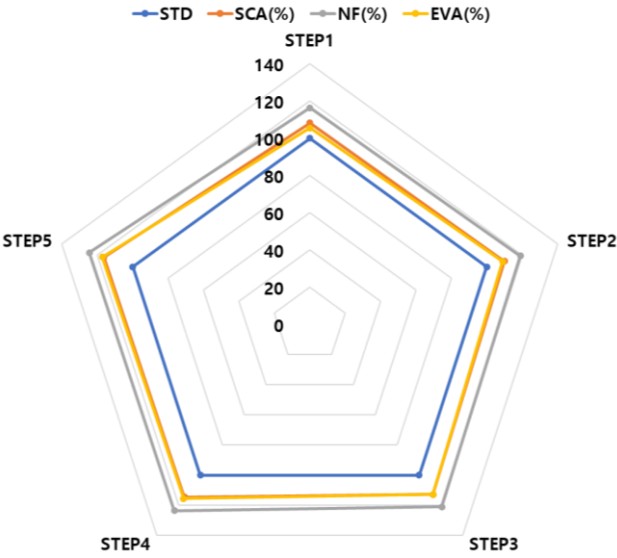

**Figure 12.** Increase/decrease rate of solid content according to the input amount of each additive.

As a result of the single additive test, the solid content, according to the addition of the additive, gradually increased as the amount of the additive increased. This is because additives, unlike materials that volatilize at high temperatures, such as water presented in the raw material, are left as residues, and thus changes occur. It was confirmed that nylon fiber, a material with large unit weight, has a relatively high solid content compared to other additives. It was confirmed that the viscosity according to the input of each additive increased by about 254% to 828% compared to the existing viscosity in the nylon fiber, and thus it was confirmed that the viscosity was high because the bonding force of the material was strong in the silane coupling agent. Overall, the change in viscosity according to the addition of additives was found to be increased.

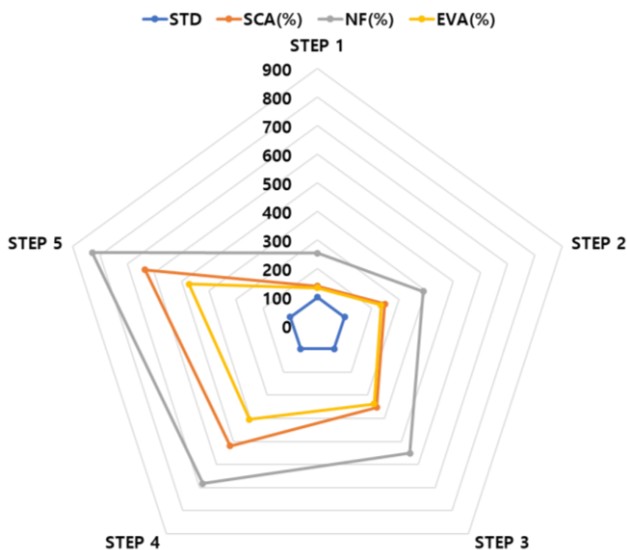

**Figure 13.** Viscosity increase/decrease rate according to the input amount of each additive.

## 4. Confirmation of the Properties of Complex Additives

### 4.1. Setting the Range of the Compound Additive's Mixing Ratio

In this study, the amount of each additive was derived according to the single additive test result in Table 10. In the case of the silane coupling agent, the increase in tensile and tear strength was gently escalated from 0.3% to 1.5%, and the decrease in elongation was also stably decreased. Accordingly, the addition ratio was set based on 1.5%, and three types of addition ratios of 1.3%, 1.4%, and 1.5% were set as a compound additive mixing range. Nylon fiber was confirmed as an excellent additive to improve the performance of tensile strength and tear strength, but it was found that the elongation rate decreased significantly due to overdosing. Accordingly, the addition ratio was set based on 2.0%, and three types of addition ratios of 1.8%, 2.0%, and 2.2% were set as a compound additive mixing range. In the case of EVA, a stable decrease in tensile and tear strength and a stable increase in elongation were observed. However, with an addition ratio of 2.0% or more, there was no difference in material properties, and only the solid content and viscosity increased. Therefore, the addition ratio was set based on 2.0%, and the addition ratio of 1.6%, 1.8%, and 2.0% was set as the compound additive mixing range.

**Table 10.** The set range of compound additives.

| Item | SCA | NF | EVA |
|---|---|---|---|
| | 1.3 | 1.8 | 1.6 |
| Additive Ratio (%) | 1.4 | 2.0 | 1.8 |
| | 1.5 | 2.2 | 2.0 |

### 4.2. Preparation of the Composite Additive's Formulation Table

A compound additive formulation table was prepared according to the formulation range set in this study, and the formulation range from 01 to 27 of the complex additive ratio (code: CA (Complex Additive Ratio)) was confirmed, as shown in Table 11 below.

### 4.3. Composite Additive Mixing Progress

A compound additive formulation table was prepared according to the formulation range set in this study and is shown in Table 12 below.

**Table 11.** Composite additive mixing range.

| | | Item | | |
| --- | --- | --- | --- | --- |
| SCA | NF | EVA | Total Additive | Test Code |
| | | 1.6 | 4.7 | CA 01 |
| | 1.8 | 1.8 | 4.9 | CA 02 |
| | | 2.0 | 5.1 | CA 03 |
| | | 1.6 | 4.9 | CA 04 |
| 1.3 | 2.0 | 1.8 | 5.1 | CA 05 |
| | | 2.0 | 5.3 | CA 06 |
| | | 1.6 | 5.1 | CA 07 |
| | 2.2 | 1.8 | 5.3 | CA 08 |
| | | 2.0 | 5.5 | CA 09 |
| | | 1.6 | 4.8 | CA 10 |
| | 1.8 | 1.8 | 5.0 | CA 11 |
| | | 2.0 | 5.2 | CA 12 |
| | | 1.6 | 5.0 | CA 13 |
| 1.4 | 2.0 | 1.8 | 5.2 | CA 14 |
| | | 2.0 | 5.4 | CA 15 |
| | | 1.6 | 5.2 | CA 16 |
| | 2.2 | 1.8 | 5.4 | CA 17 |
| | | 2.0 | 5.6 | CA 18 |
| | | 1.6 | 4.9 | CA 19 |
| | 1.8 | 1.8 | 5.1 | CA 20 |
| | | 2.0 | 5.3 | CA 21 |
| | | 1.6 | 5.1 | CA 22 |
| 1.5 | 2.0 | 1.8 | 5.3 | CA 23 |
| | | 2.0 | 5.5 | CA 24 |
| | | 1.6 | 5.3 | CA 25 |
| | 2.2 | 1.8 | 5.5 | CA 26 |
| | | 2.0 | 5.7 | CA 27 |

*4.4. Composite Additive Test Results*

4.4.1. Confirmation of Items Excluded from Evaluation According to the State of the Coating Film

The purpose of compounding materials is to supplement their shortcomings and to derive optimal physical properties when using additives with different properties. However, since the coefficients of expansion with respect to volume or heat are opposite to each other depending on the type of additives used, even if a homogeneous mixture is created, non-uniformity of the coating film is caused due to drying and hardening of the material. Therefore, test evaluation was excluded for heterogeneous materials as follows from a total of 27 test and evaluation materials.

(1) EVA 2.0% blending ratio

In the case of CA 03, CA 06, CA 09, CA 12, CA 15, CA 18, CA 21, CA 24, and CA 27 specimens with more than 2.0% EVA added during the manufacturing process of the composite additive specimen, sufficient mixing was achieved, as shown in Photo 3 below. Even after that, residual EVA was precipitated. It was judged that EVA added together with other additives in the material was not mixed in the material and precipitated. Therefore, in this study, this test was used to prevent the deterioration of the properties of the waterproofing layer due to material precipitation and to secure a homogeneous waterproofing performance.

(2) Nylon fiber 2.2% mixing ratio

In the test specimens of CA 07 ~ CA 09, CA 16 ~ CA 18, and CA 25 ~ CA 27, in which 2.2% or more of nylon fiber was added during the manufacturing process of the composite additive specimen, the aggregation of the material indicating poor workability and the resulting imbalance in the coating film thickness were partially due to the mixing ratio. The specimens in which material agglomeration occurred were as follows; CA 08 ~ CA 09,

CA 17 ~ CA 18, and CA 25 ~ CA 27. Furthermore, the agglomeration phenomenon was clearly observed at a viscosity of 4500 mPa·s or higher, and the viscosity of 5000 mPa·s or higher in the future was excluded from the formulation. Accordingly, the state of the poured material after mixing is shown in the following Figure 14.

**Table 12.** Complex additive formulation process.

| Mixing Procedure | Mixing Procedure |
| --- | --- |
| Content | Content |

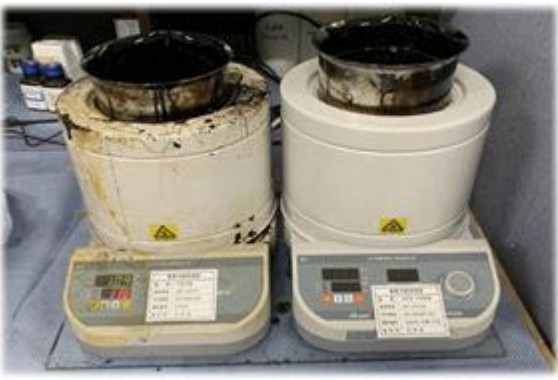

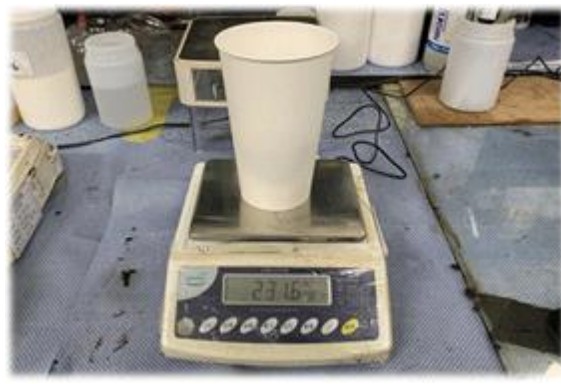

1. Melting of water-soluble rubber asphalt (raw material)
- Asphalt has excellent temperature sensitivity and hardens at low temperatures or increases in viscosity.

2. Weighing of water-soluble rubber asphalt (raw material)
- For accurate weighing of additives, weighing of raw materials precedes.

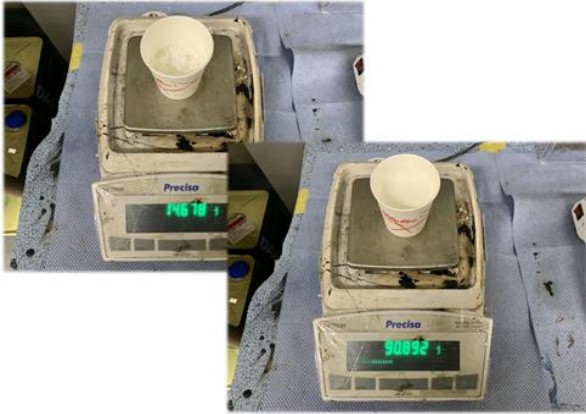

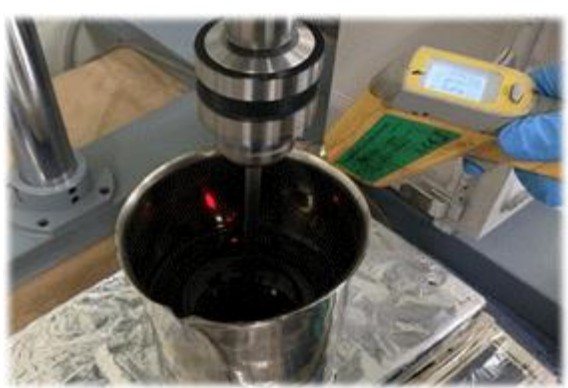

3. Additive Weighing
- Measure each additive according to the raw material weighing ratio.

Mixing temperature control
- Prevention of non-homogeneous mixing, such as material aggregation and deflection.

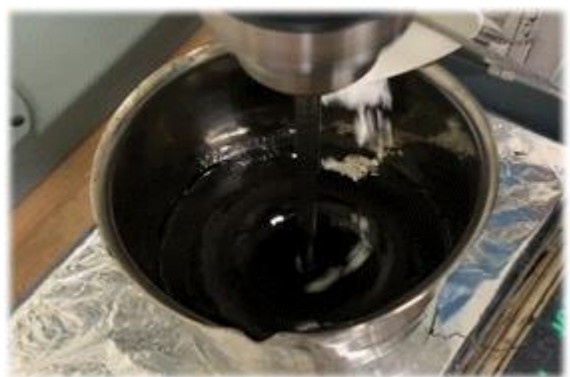

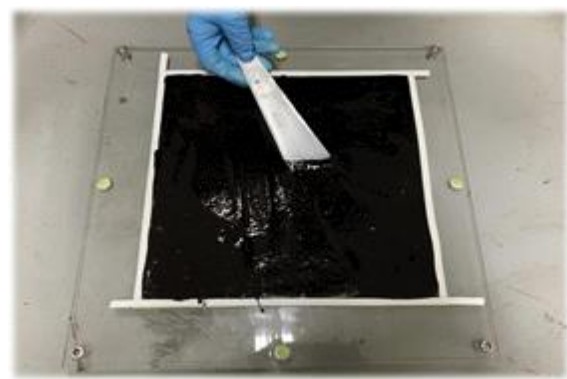

Additive input
- Induces homogeneous mixing of asphalt and additives at high temperatures.

Material application
- Apply the blended material to the acrylic plate equipped with a level so that the thickness of the material is not deflected.

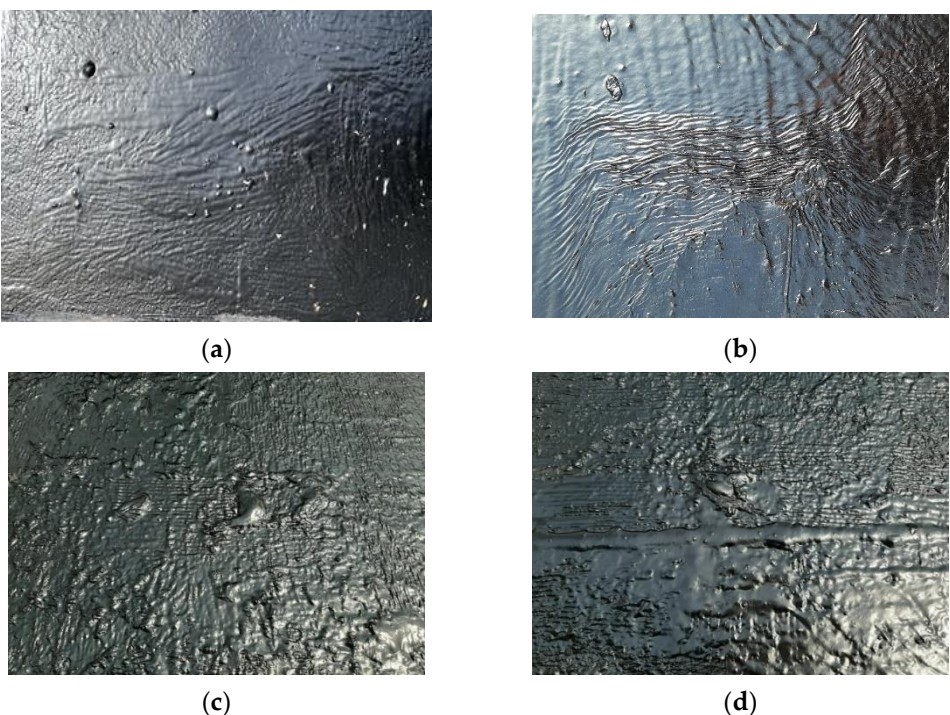

**Figure 14.** Mixing error in some items in nylon fiber 2.2% blend ratio. (**a**) Film thickness imbalance; (**b**) poor workability; (**c**) agglomeration of materials; (**d**) agglomeration of materials.

(3) Checking the hardened state of the normal coating film waterproofing material
Specimens in the normal hardened state except for material separation and poor workability in the coating film curing process included CA 01 ~ CA 02, CA 04 ~ CA 05, CA 07, CA 10 ~ CA 11, CA 13 ~ CA 14, CA 16, CA 19 ~ CA 20, and CA 22 ~ CA 23. Physical property tests were performed on a total of 14 items, and the hardened state of a normal specimen is shown in Figure 15 below.

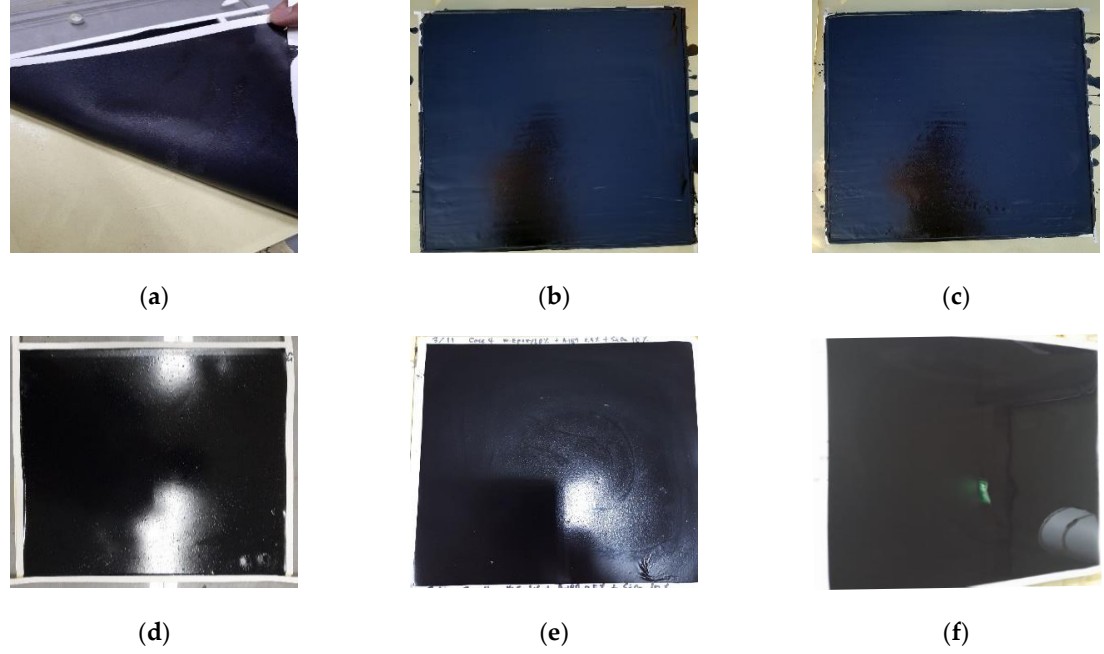

**Figure 15.** Hardened state of material in steady state. (**a**) Checking material separation; (**b**) steady-state coating; (**c**) steady-state coating; (**d**) steady-state coating; (**e**) steady-state coating; (**f**) steady-state coating.

#### 4.4.2. Physical Property Test Results

Table 13 shows the results of checking the properties of the water-soluble rubber-asphalt-based waterproofing material according to the additive compound mixture. According to the initial formulation design, the formulation of 27 material groups was carried out, but test specimens with poor coating conditions, such as those with material separation, were excluded. They were excluded because the occurrence rate of defects in the curing process of the material increases when the amount of addition is excessive. When the combination of the silane coupling agent, which improves the strength of the material, and the nylon fiber and EVA, which improves the elongation, is biased, high tensile strength or high tensile strength is excluded. Due to the fact that the silane coupling agent maintains the elongation rate, it is considered that it is not suitable for the quality standards of exposed waterproofing materials being used for construction. The degree of physical property improvement of the formulation design used in this study was confirmed by the mutual analysis of each test item evaluated and quality standards.

**Table 13.** Single mixture ratio additive properties.

| Item (Code) | Tensile Strength (N/mm$^2$) | Elongation (%) | Test Results Tear Resistance (N/mm) | Solidity Ratio (%) | Viscosity (mPa·s) | Item (Code) | Tensile Strength (N/mm$^2$) | Elongation (%) | Test Results Tear Resistance (N/mm) | Solidity Ratio (%) | Viscosity (mPa·s) |
|---|---|---|---|---|---|---|---|---|---|---|---|
| STD | 0.33 | 1352 | 3.21 | 73.33 | 522 | | | | - | | |
| CA 01 | 1.89 | 721 | 9.78 | 88.67 | 3480 | CA 13 | 2.45 | 449 | 14.65 | 90.00 | 3947 |
| CA 02 | 1.82 | 767 | 9.10 | 88.67 | 3665 | CA 14 | 2.32 | 478 | 12.78 | 91.33 | 4443 |
| CA 04 | 2.21 | 597 | 11.51 | 89.33 | 3761 | CA 16 | 2.61 | 389 | 14.88 | 92.67 | 4511 |
| CA 05 | 2.04 | 752 | 10.68 | 90.67 | 4291 | CA 19 | 2.22 | 536 | 12.14 | 89.33 | 3714 |
| CA 07 | 2.34 | 451 | 13.68 | 92.67 | 4415 | CA 20 | 2.08 | 513 | 10.98 | 90.67 | 4139 |
| CA 10 | 2.10 | 649 | 11.12 | 88.67 | 3574 | CA 22 | 2.56 | 402 | 15.24 | 91.33 | 4253 |
| CA 11 | 1.95 | 579 | 10.11 | 90.00 | 3916 | CA 23 | 2.47 | 432 | 14.12 | 92.00 | 4331 |

#### 4.5. Analysis of the Composite Additive's Test Results

#### 4.5.1. Tensile Strength

A comparison of the test results and quality standards for the overall tensile strength among the composite additive test results is shown in Figure 16

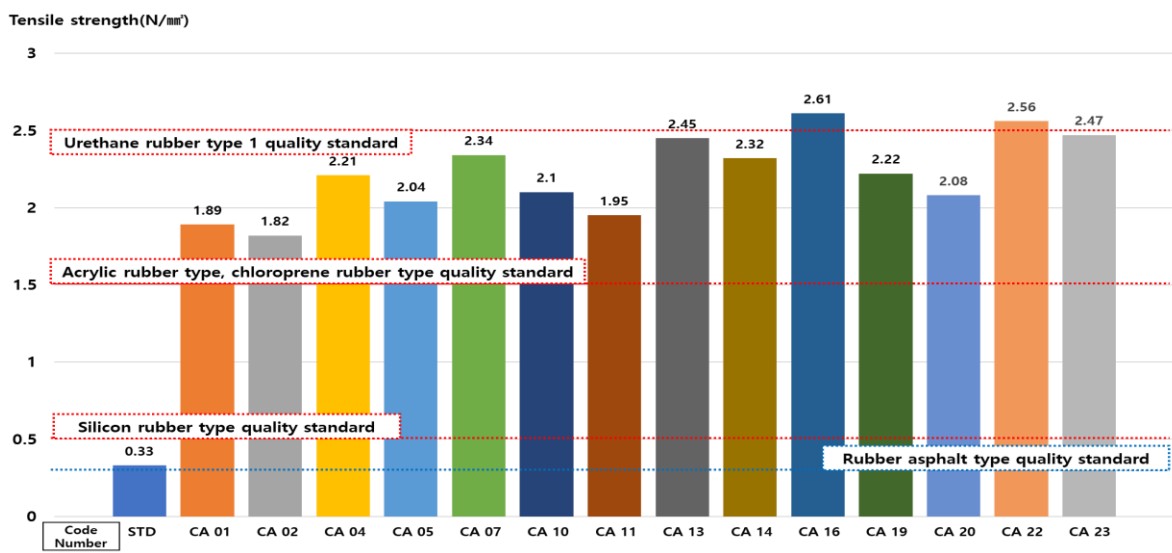

**Figure 16.** Comparison of composite additive tensile strength test results and quality standards.

As a result of applying the composite additive to the rubber-asphalt-based waterproofing material used in this study, the minimum tensile strength was increased by about 552% to 1.82 N/mm$^2$, and the maximum tensile strength was increased by about 791% to 2.61 N/mm$^2$. This is a satisfactory value exceeding the quality standards of acrylic

rubber, chloroprene rubber, and silicone rubber of Korean Industrial Standard KS F 3211. According to the amount of silane coupling agent and nylon fiber added to the water-soluble rubber-asphalt-based waterproofing material, it was found that the maximum tensile strength quality standard presented in the Korean Industrial Standard KS F 3211 was satisfied.

### 4.5.2. Elongation

The comparison of the test results and quality standards for the overall elongation among the test results of the complex additive is shown in Figure 17.

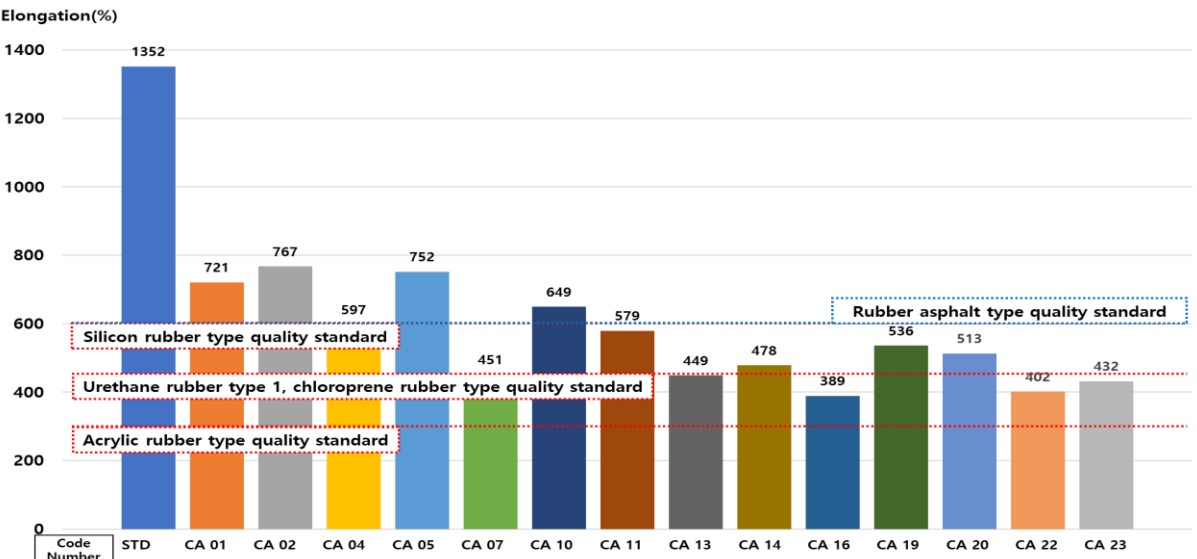

**Figure 17.** Comparison of composite additive elongation test results and quality standards.

As a result of applying the composite additive to the rubber-asphalt-based waterproofing material used in this study, the minimum elongation was reduced by about 963% to 389%, and the maximum elongation was reduced by about 585% to 767%. This is a value that exceeds the quality standards for the elongation of acrylic rubber of KS F 3211 of the Korean industry standard, but in the case of CA 13, CA 16, CA 22, and CA 23, it did not satisfy the quality standards for urethane rubber type 1 and chloroprene rubber type, and CA 01, CA 02, CA 05, and CA 10 were found to not satisfy the rubber-asphalt-based quality standards and silicone-rubber-based quality standards, which are the original quality standards of the material, in other additive mixing ratios.

### 4.5.3. Tear Strength

A comparison of the test results and quality standards for the overall tear strength among the composite additive test results is shown in Figure 18.

As a result of applying the composite additive to the rubber-asphalt-based waterproofing material used in this study, the minimum tearing strength was increased by about 284% to 9.10 N/mm$^2$, and the maximum tearing strength was increased by about 475% to 15.24 N/mm$^2$. This is a satisfactory value that exceeds the quality standards of acrylic rubber and silicone rubber of Korean Industrial Standard KS F 3211, and in some items, such as CA 16 and CA 22, satisfactory results exceeded the quality standards of first-class urethane rubber and chloroprene rubber. According to the amount of silane coupling agent and nylon fiber added to the water-soluble rubber-asphalt-based waterproofing material, it was found that the maximum tensile strength quality standard presented in the Korean Industrial Standard KS F 3211 was satisfied.

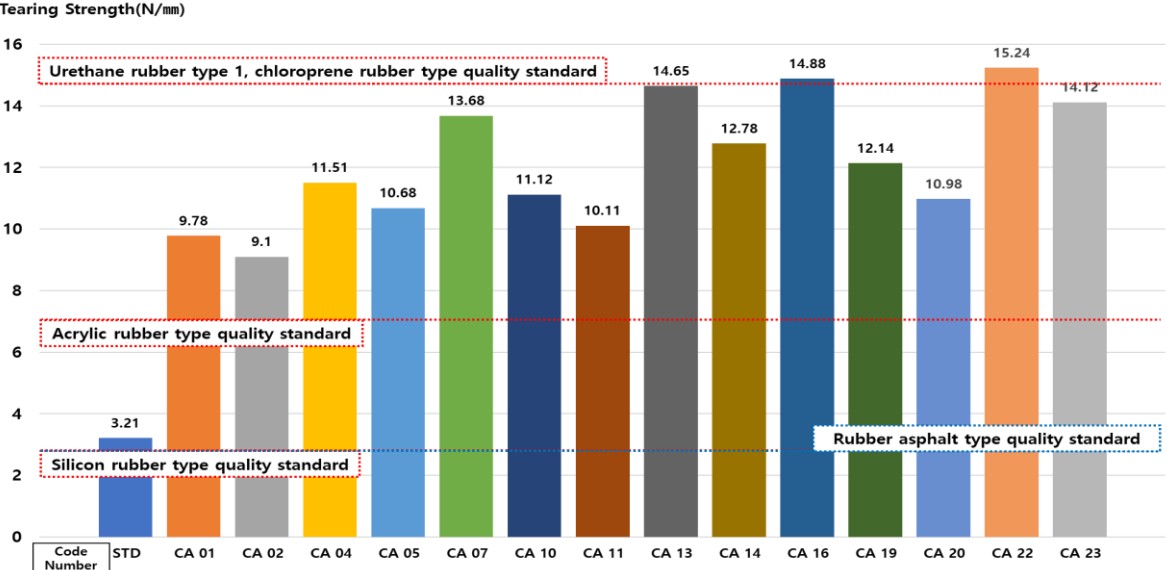

**Figure 18.** Comparison of composite additive tear strength test results and quality standards.

4.5.4. Solids

The test results for the total solid content in the composite additive test results are shown in Figure 19.

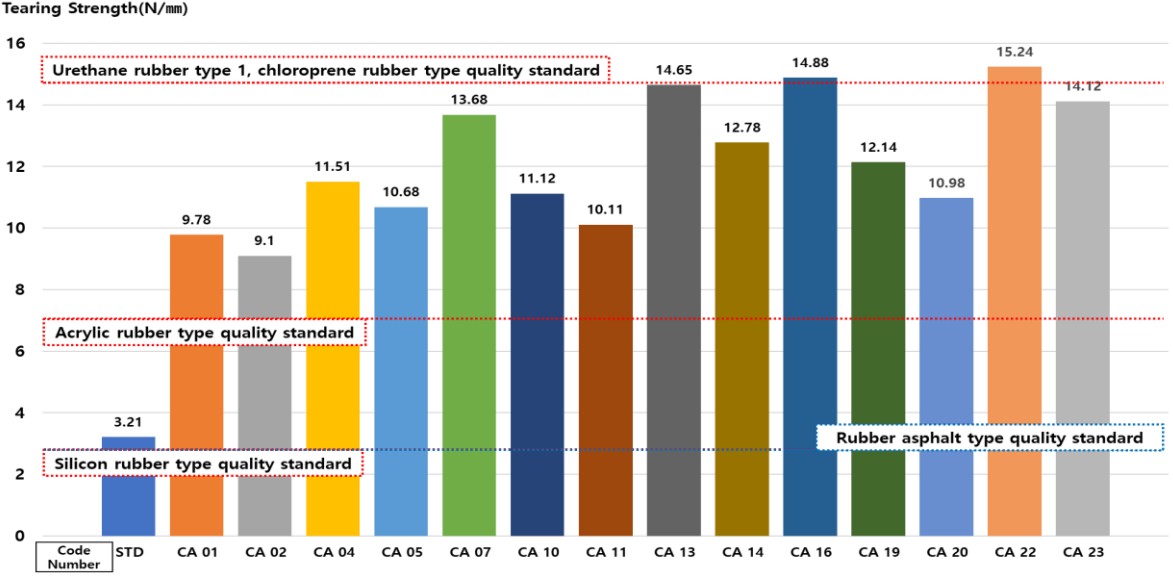

**Figure 19.** Composite additive solid content test result.

As a result of applying the composite additive to the rubber-asphalt-based waterproofing material used in this study, the minimum solid content was increased by about 121% to 88.67%, and the maximum solid content was increased by about 126% to 92.67%. There is no separate quality standard for these solids, but since it is possible to predict the degree of decrease in the thickness of the waterproofing layer due to hardening after the on-site casting of the waterproofing material, the mixing ratio with a solid content of 90% or more is CA 05, CA 07, CA 11, CA 13, and CA 14, as well as CA 16, CA 20, CA 21, CA 23, etc., and it was confirmed that 90% or more of the solid content was expressed at 5.0% or more of the total mixing ratio.

#### 4.5.5. Viscosity

Among the composite additive test results, the test results for the entire viscosity are shown in Figure 20.

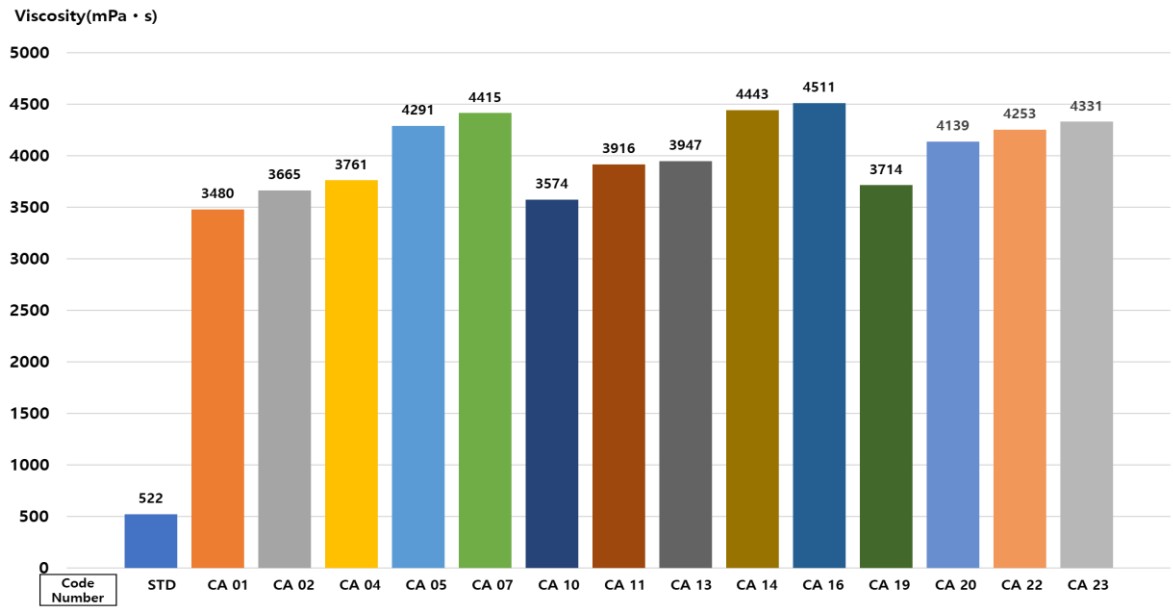

**Figure 20.** Composite additive viscosity test results.

As a result of applying the composite additive to the rubber-asphalt-based waterproofing material used in this study, the lowest viscosity was increased by about 667% to 3480 mPa·s, and the maximum viscosity was increased by about 864% to 4511 mPa·s. There is no separate quality standard for this viscosity, but when the waterproofing material is installed on-site, the construction of the waterproofing coating film with an appropriate viscosity causes material agglomeration, making it impossible for the installer to construct a waterproofing layer with a homogeneous thickness. Cracks were found at about 4000 mPa·s, which is superior workability than that previously suggested of about 4500 mPa·s for the compounding ratio of CA 05, CA 07, CA 14, CA 16, CA 20, CA 22, CA 23, etc. There is some difficulty, however, and in particular, the mixing ratio of CA 07, CA 14, and CA 16 requires attention during the construction process.

#### 5. Conclusions

The conclusion of this 'Study on changes in physical properties due to the use of additives (silane coupling agent, nylon fiber, and EVA) to improve the basic properties of water-soluble rubber-asphalt-based paint film waterproofing material for indoor use in building structures' is as follows.

Tensile strength and tear strength test results for CA 16 and CA 22 test specimens were determined according to the quality standards for tensile strength of 2.5 N/mm$^2$ and tear strength, which is the highest quality standard for urethane rubber class 1 among the quality standards presented in Korean Industrial Standard KS F 3211. The quality standard of 14.7 N/mm was exceeded, and this is because the water-soluble rubber asphalt coating waterproofing material, which was only applied to the inside of the structure, was limitedly applied to the inside of the structure due to its low tensile strength. It is judged that excellent physical properties, such as preventing the tearing of the waterproofing layer due to cracks in the structure, are possible.

As a result of the elongation test result, it was found that the high tensile properties of the water-soluble rubber-asphalt-based waterproofing materials were rapidly reduced by using the additives. According to the use of complex additives, all specimens except CA 01,

CA 02, CA 05, and CA 10 did not satisfy the rubber asphalt quality standard, which is the quality standard of raw materials. Therefore, it is judged that it is necessary to adjust the elongation rate using the above compounding ratio.

According to the results of the solid content and viscosity test, it was possible to improve the flowability of the material by more than 90% due to a reduction in thickness after construction and liquefaction, which is the limit of water-soluble rubber asphalt coating waterproofing materials. In particular, in the case of viscosity, if the formulation design is made within a viscosity of about 4500 mPa·s, it is judged that it is possible to secure a homogeneous waterproofing layer.

Summarizing the comprehensive conclusions confirmed through this study, it was confirmed that the quality improvement of materials was excellent according to the use of additives. However, as there are some incomplete items in some quality standards, it is considered that the quality of materials needs to be improved a little more. Therefore, it has been determined through this study that if additional research and development based on the test specimen mixing ratio of CA 04, CA 05, CA 10, CA 13, and CA 19 is conducted based on the data of this study, it would be possible to develop a stable water-soluble rubber-asphalt-based waterproofing material that exceeds quality standards.

**Author Contributions:** Conceptualization, S.-h.Y., N.-h.H., K.-w.A. and S.-k.O.; methodology, S.-h.Y., N.-h.H. and K.-w.A.; software, experimental plan, K.-w.A. and S.-k.O.; formal analysis, investigation, resources, N.-h.H.; data curation, S.-h.Y. and K.-w.A.; writing—original draft preparation, S.-h.Y. and N.-h.H.; writing—review and editing, S.-k.O. All authors have read and agreed to the published version of the manuscript.

**Funding:** This research received no external funding.

**Institutional Review Board Statement:** Not applicable.

**Informed Consent Statement:** Not applicable.

**Data Availability Statement:** Not applicable.

**Conflicts of Interest:** The authors declare no conflict of interest. The founding sponsors had no role in the design of the study; in the collection, analyses, or interpretation of data; in the writing of the manuscript, and in the decision to publish the results.

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
