# Peer review of "A Study on the Change in Properties by Using an Additive with Water-Soluble Rubber-Asphalt-Based Waterproof Coating Materials"

_applsci, doi:10.3390/app12094599_

Round 1

Reviewer 1 Report

The article is a comprehensive scientific study with practical significance.

However, the aspect of the economic efficiency of introducing additives into the asphalt mixture remained undisclosed.

How much more expensive is the proposed coatings compared to analogues?

How much more durable are they?

These points, in my opinion, need clarification.

Author Response

The article is a comprehensive scientific study with practical significance.

However, the aspect of the economic efficiency of introducing additives into the asphalt mixture remained undisclosed.

Q.1 How much more expensive is the proposed coatings compared to analogues?

A.1 This study is intended to be an experiment designed to determine the degree of improvement in the performance of waterproofing coatings. Accurate data on the cost difference for similar products is likely to be possible after trial mass production is conducted and the relevant costs are calculated thereafter. However, based on the current estimations by the manufacturers, it is expected to that the costs will be cheaper by about 10-20% or more compared to the conventional polyurethane coating waterproofing materials that are widely used.

Q.2 How much more durable are they?

A.2 A comparison of overall durability can be determined only after the test and evaluation of all items presented in Korean Industrial Standard KS F 3211 is carried out. It was confirmed that the tensile strength was less than the quality standard by about 24%, and it was confirmed that the performance improved by about 500% or more compared to the waterproofing material of the same asphalt series.

Q.3 These points, in my opinion, need clarification.

A.3 On the topic of economic applicability, the purpose of the study is to illustrate that this study is primarily intended to compare the degree of improvement in physical properties according to additives. Additives were selected and research was conducted within the range that did not exceed the production cost of other waterproofing materials (about 80-90%).

Reviewer 2 Report

  • Abstract should be added with some conclusion-based sentences.
  • The introduction should be reorganized with some citations surrounding the topic.
  • All of the results are not associated with reasons provided.
  • The conclusions are too long to catch the viewpoints.
  • Double check the expressions throughout the manuscript.
  • The discussion section needs to add more descriptions to strengthen strategies statements, and associated reasons should also be given.
  • Most of the references are old. Some recent studies are suggested added.

Author Response

Q.1 Abstract should be added with some conclusion-based sentences.

A.1 In the abstract section, the following has been added; “Therefore, in this study, a total of 27 complex additive tests were conducted, and among them, basic property evaluation was performed on 14 items except for 13 items that had defects in the drying process. As a result, in the test conducted with complex additives, about 98% to 104% of the quality standard was obtained, and it is judged that stable waterproofing performance will appear if additional quality improvement is made based on this combination in the future.”

Q.2 The introduction should be reorganized with some citations surrounding the topic.

A.2 Citations have been included in the introduction of the article.

Q.3 All of the results are not associated with reasons provided.

A.3 This study is an experimental article that discusses the improvement of the low tensile properties and high elongation of water-soluble rubber asphalt-based waterproofing materials based on the use of additives. The degree of improvement was analyzed by comparing the performance results in compliance to the Korean industry standard quality standards according to the purpose of the study. This is further discussed in detail in the revised conclusion section of the article.

Q.4 The conclusions are too long to catch the viewpoints.

A.4 The conclusion part was simplified into three parts, and a comprehensive conclusion sentence was added accordingly to make it easier to grasp the point of view.

Q.5 Double check the expressions throughout the manuscript.

A.5 The template and numbering system in the manuscript were checked again and corrected.

Q.6 The discussion section needs to add more descriptions to strengthen strategies statements, and associated reasons should also be given.

A.6 In order to strengthen the strategic statements in the conclusion section and improve the semantics, the following section was included; “Summarizing the comprehensive conclusions confirmed through this study, it was determined that the quality improvement of materials was confirmed by the usage of additives. However, as there are some incomplete items in some quality standards, it is considered that the quality of materials needs to be improved further. Therefore, if additional research and development based on the test specimen mixing ratio of CA 04, CA 05, CA 10, CA 13, and CA 19 is conducted based on the data of this study, it is possible to develop a stable water-soluble rubber asphalt-based waterproofing material that exceeds the quality standards in the future.”

Q.7 Most of the references are old. Some recent studies are suggested added.

A.7 References [19], [20], and [21] referring to recent studies were added to the references section.

Round 2

Reviewer 1 Report

Corrections and explanations accepted.